# Single-Nucleotide Polymorphism on Spermatogenesis Associated 16 Gene-Coding Region Affecting Bovine Leukemia Virus Proviral Load

**DOI:** 10.3390/vetsci9060275

**Published:** 2022-06-06

**Authors:** Hirohisa Mekata, Mari Yamamoto

**Affiliations:** Center for Animal Disease Control, University of Miyazaki, 1-1 Gakuen-Kibanadai-Nishi, Miyazaki 8892192, Japan; markoba@cc.miyazaki-u.ac.jp

**Keywords:** alleles, bovine leukemia virus, cattle, enzootic bovine leukosis, genome-wide association study, single-nucleotide polymorphism

## Abstract

Bovine leukemia virus (BLV) is an etiological agent of malignant lymphoma in cattle and is endemic in many cattle-breeding countries. Thus, the development of cattle genetically resistant to BLV is desirable. The purpose of this study was to identify novel single-nucleotide polymorphisms (SNPs) related to resistance to BLV. A total of 146 DNA samples from cattle with high BLV proviral loads (PVLs) and 142 samples from cattle with low PVLs were used for a genome-wide association study (GWAS). For the verification of the GWAS results, an additional 1342 and 456 DNA samples from BLV-infected Japanese Black and Holstein cattle, respectively, were used for an SNP genotyping PCR to compare the genotypes for the identified SNPs and PVLs. An SNP located on the spermatogenesis associated 16 (SPATA16)-coding region on bovine chromosome 1 was found to exceed the moderate threshold (*p* < 1.0 × 10^−5^) in the Additive and Dominant models of the GWAS. The SNP genotyping PCR revealed that the median values of the PVL were 1278 copies/50 ng of genomic DNA for the major homozygous, 843 for the heterozygous, and 621 for the minor homozygous genotypes in the Japanese Black cattle (*p* < 0.0001). A similar tendency was also observed in the Holstein cattle. We found that cattle with the minor allele for this SNP showed 20–25% lower PVLs. Although the mechanisms through which this SNP impacts the PVL remain unknown, we found a novel SNP related to BLV resistance located on the SPATA16 gene-coding region on bovine chromosome 1.

## 1. Introduction

Bovine leukemia virus (BLV) is an etiological agent of malignant lymphoma in cattle and water buffaloes. This virus is classified into the *Retroviridae* family and the *Deltaretrovirus* genus, and the disease is classed as an enzootic bovine leukosis (EBL). Although most BLV-infected cattle remain EBL-free for life, 1–5% develop EBL a few years after infection [1,2]. Australia, New Zealand, and many Western European countries have successfully eliminated BLV through national eradication programs. However, many other countries have not performed national eradication programs, and high seroprevalences have been confirmed in countries such as Brazil, Japan, and the USA [3,4,5]. Although many workers involved in the cattle industry in Japan have worked to control BLV, the reported number of cattle with bovine leukosis is increasing every year. Thus, the establishment of a novel approach to control BLV is desirable.

An animal’s BLV proviral load (PVL) fluctuates months after infection and becomes stable over time [6]. Stable-state PVLs vary widely between individual cattle and can range from fewer than 2 copies per 50 ng of genomic DNA to more than 2000. Some recent studies have reported that cattle with higher PVLs have a higher risk of viral transmission and EBL development [7,8,9]. In addition, BLV infection induces the exhaustion of the immune system, and cattle with high PVLs tend to show a decline in productivity, such as a reductions in carcass weight, or severe clinical mastitis [10,11,12]. Therefore, reducing the number of cattle with high PVLs is a key factor for reducing BLV infections, EBL development, and productivity losses.

An animal’s PVL is thought to be under the control of multiple host genes. The bovine leukocyte antigen (BoLA) class II gene, which is the major histocompatibility complex class II system in cattle, is an especially well-known factor strongly associated with PVL [13,14,15,16,17,18]. BoLA class II is a highly polymorphic gene set located on bovine chromosome 23. This gene set is responsible for the presentation of the antigenic peptide to immune cells and is associated with genetic resistance against many infectious diseases [19]. Further studies are still required to understand whether the affinity between antigenic peptides and the binding pocket of immune cells could affect BLV resistance [20]. Although increasing the number of cattle with BLV-resistant BoLA class II alleles could be useful for BLV control, the deviation of specific alleles would be lost due to genetic diversity and lead to higher sensitivity against other bovine infectious diseases. Thus, we need to find various BLV-resistant genes beyond the BoLA gene set. A genome-wide association study (GWAS) using a single-nucleotide polymorphism (SNP) array is one of the most common methods for identifying novel genes related to susceptibility or resistance to infectious diseases. To the best of our knowledge, four GWAS studies have been conducted with BLV [21,22,23,24]. These studies revealed that the BoLA gene set comprised the most significant SNPs related to BLV infection. On the other hand, no SNP related to BLV resistance outside chromosome 23 has been identified. The purpose of this study was to identify novel candidate SNPs related to BLV resistance.

## 2. Materials and Methods

### 2.1. Samples

All the blood samples used in this study were obtained from Japanese Black and Holstein cattle (female, >12 months old) in the Mie, Oita, Kumamoto, Miyazaki, and Kagoshima prefectures in Japan from January 2017 to March 2022 by local clinical veterinarians for BLV infection diagnosis. All the blood samples were confirmed positive for anti-BLV antibodies and BLV proviral DNA with an enzyme-linked immunosorbent assay (ELISA) and real-time PCR, respectively. The ELISA, extraction of genomic DNA, and real-time PCR are described in detail in our previous report [25].

### 2.2. GWAS

We previously reported that cattle with fewer than 100 BLV proviral copies/50 ng of genomic DNA were hardly able to transmit the virus to BLV-free cattle [25,26]. On the other hand, cattle with more than 2000 copies/50 ng could easily vertically transmit the virus to their calves [7]. Thus, we classified cattle with more than 2000 and fewer than 100 copies/50 ng as high and low PVL groups, respectively. A total of 288 DNA samples (high PVL group: 146; low PVL group: 142) from the Japanese Black cattle were used for SNP arrays. The samples were collected from 149 beef cattle production farms, and a maximum of 3 samples per farm were used for maintaining the pedigree diversity of the analyzed samples. The Axiom Bovine Genotyping v3 Array (Thermo Fisher Scientific, Waltham, MA, USA) comprising probes targeting 65,003 SNPs and the Affymetrix GeneTitan system (Thermo Fisher Scientific) were used for the SNP arrays. The SNP arrays were performed by TaKaRa Bio (Kusatsu, Japan) and Thermo Fisher Scientific Japan Group (Tokyo, Japan). All the genetic association analyses were performed with the SNP & Variation Suite version 8.9.1 software (Golden Helix, Bozeman, MT, USA). The Bos taurus UMD3.1 in the NCBI database and Axiom BovMDv3 provided by Thermo Fisher Scientific were used as the genome reference and the genetic marker map, respectively. A total of 61,754 SNPs were successfully mapped to the 29 autosomal chromosomes of cattle. To control the quality of the analysis, samples with call rates less than 0.95 and SNPs with call rates less than 0.95, minor allele frequencies less than 0.01, and Hardy–Weinberg equilibria less than 0.001 were excluded. In addition, linkage disequilibrium (LD) pruning was performed to remove linked SNPs (r^2^ > 0.9). The Additive, Dominant, and Recessive models based on associations of the SNPs with case (low PVL) and control (high PVL) were analyzed using Fisher’s exact test. The genomic inflation factor (λ) was calculated in each model, and the genomic control method was used to control sample stratification [27]. The genome-wide significance threshold was set based on the Bonferroni correction as *p* = 1.449 × 10^−6^ (0.05/34,491) because 34,491 SNPs passed the quality control filters and LD pruning. Additionally, a moderate threshold (*p* = 1.0 × 10^−5^) was set because the *p*-value controlled by the genomic control method is thought to be conservative [28].

### 2.3. rhAmp SNP Genotyping

rhAmp SNP genotyping was used for a real-time PCR-based SNP genotyping assay [29]. A total of 1798 (Japanese Black cattle: 1342; Holstein: 456) DNA samples from BLV-infected cattle were used for the assay. The median (mean) values of the PVLs in Japanese Black cattle were 1042 (1529) copies/50 ng including 408 cattle of high and 246 cattle of low PVL groups. On the other hand, the median (mean) values of the PVLs in Holstein cattle were 2399 (3468) copies/50 ng including 249 cattle of high and 99 cattle of low PVL groups. The complete genome of Bos taurus UMD3.1 chromosome 1 (Accession No. GK000001.2) was obtained from GenBank. Nucleotide sequences, including those both 500 bp upstream and downstream from the SNP identified by the GWAS, were extracted. The candidates of primer sets were designed based on the extracted sequence using the rhAmp Genotyping Design Tool (Integrated DNA Technologies, Coralville, IA, USA). After the verification of the candidate primer sets, we used the following primer sets—allele-specific primer: 5′-CTCTGTTTCACCACCAC[A/G]ACTGT-3′; locus-specific reverse primer: 5′-GCCATGCAGCCTATATAAGTGATGTTCTTGT-3′. This primer set was included in the custom rhAmp SNP assays (Integrated DNA Technologies). rhAmp SNP genotyping was performed using the genomic DNA mixed with rhAmp Genotyping Master Mix, rhAmp Reporter Mix (Integrated DNA Technologies), and the custom rhAmp SNP assays following the manufacturer’s instructions. PCR amplification was performed as follows: an initial PCR activation step at 95 °C for 10 min, and then 40 cycles of 95 °C for 10 s, 60 °C for 30 s, and 68 °C for 20 s. A major allele of A was detected with the FAM channel, and a minor allele of G was detected with the VIC channel.

### 2.4. Statistical Analysis

The chi-square test was used for comparisons between the coloration of PVL groups and each genotype. The D’Agostino–Pearson normality test was used to verify the normal and lognormal PVL distribution. Because the PVL was not distributed normally or lognormally, Kruskal–Wallis and Dunn’s multiple-comparison tests were used for comparisons among the PVLs and each genotype. These analyses were performed using the GraphPad Prism 6 software (GraphPad Software, La Jolla, CA, USA). *p* < 0.05 was considered statistically significant.

## 3. Results

In this study, we used 146 samples from Japanese Black cattle with high PVLs and 142 from cattle with low PVLs for the Axiom Bovine Genotyping v3 Array. No samples and 25,434 SNPs were filtered out by the quality control process. After the LD pruning, 34,491 SNPs were retained for downstream analyses. The genomic inflation factors (Additive model: λ = 1.20; Dominant model: λ = 1.16; Recessive model: λ = 1.31) and the quantile–quantile (QQ) plots indicated the presence of systematic bias (Figure 1a). Thus, the genomic control method was used to control the sample stratification, and the systemic bias was controlled (Figure 1b). The controlled *p*-values are represented in Manhattan plots (Figure 2). No SNPs exceeded the genome-wide significance threshold in all the models. One SNP located on chromosome 1 at 95,193,165 (SNP ID: AX-168291905) and three SNPs located on chromosome 23 at 25,642,674 (AX-124375550), 28,568,745 (AX-185116447), and 30,773,701 (AX-115113358) exceeded the moderate threshold in the Additive model (Figure 2a and Table 1). AX-168291905 also exceeded the moderate threshold in the Dominant model (Figure 2b). On the other hand, no SNPs exceeded the moderate threshold in the Recessive model (Figure 2c). Previous studies have shown that most of the significant SNPs related to BLV infection are located on chromosome 23. Thus, we focused on the SNP on chromosome 1 (AX-168291905). This region encodes spermatogenesis associated 16 (SPATA16: Chr1; 94,965,198-95,364,350), and the SNP is located between exons 4 and 5 of the SPATA16 gene. In addition, this SNP is close to the coding region of glutathione peroxidase 6 (GPX6: Chr1;95,159,026-95,166,199), the zinc finger, and the SCAN domain containing 23 (ZSCAN23: Chr1;95,203,732-95,206,770).

To follow up on the result of the GWAS, we investigated the correlation between the PVL and the genotypes of the identified SNPs with an additional 1798 DNA samples using the rhAmp SNP genotyping system. This system discriminated 95 samples of the major (A_A) and minor (G_G) homozygous genotypes and the heterozygous (A_G) genotype in the identified SNP by one real-time PCR run (Figure 3). In the high-PVL group, 4.7% (19/408) of Japanese Black and 4.4% (11/249) of Holstein cattle had the minor homozygous genotype, whereas approximately twice this number of Japanese Black, 9.8% (24/246), and Holstein cattle, 9.1% (9/99), had this genotype in the low PVL group (Table 2). In a total of 1342 samples of BLV-infected Japanese Black cattle, the median (mean) values of the PVL were 1278 (1757) copies/50 ng for the major homozygous genotype, 843 (1307) copies/50 ng for the heterozygous genotype, and 621 (1015) copies/50 ng for the minor homozygous genotype (Figure 4a). Significant differences between the major homozygous and the heterozygous genotypes (*p* < 0.0001) and between the major and minor homozygous genotypes (*p* = 0.0003) were confirmed. On the other hand, in a total of 456 samples of BLV-infected Holstein cattle, the median (mean) values of the proviral loads were 2896 (3799) copies/50 ng for the major homozygous genotype, 2033 (3153) copies/50 ng for the heterozygous genotype, and 388 (2101) copies/50 ng for the minor homozygous genotype (Figure 4b). A significant difference between the major and minor homozygous genotypes was confirmed (*p* = 0.048). Of the Japanese Black and Holstein cattle, only 7.2% (97/1342) and 5.2% (24/456) had the minor homozygous genotype, respectively.

## 4. Discussion

First, we performed a GWAS between cattle with high and low PVLs. The early deviation of the *p*-values from the expected *p*-values in the QQ plots suggested sample stratifications (Figure 1a). Thus, we used the genomic control method to control these stratifications (Figure 1b). The principal component analysis (PCA) method and mixed-model approach are preferred for use with a GWAS because the *p*-values modified by the genomic control method are thought to be conservative [28]. However, the early separation in the QQ plots was not improved by the PCA method and mixed-model approach (data not shown). Therefore, we selected the genomic control method. Although the samples were selected from 149 beef cattle production farms in the Kumamoto, Kagoshima, and Miyazaki prefectures, the pedigree of the sampling cattle might have been biased because a limited number of stud bulls have been managed in each prefecture for the Japanese Black cattle. This might have affected the sample stratification. In this study, only an SNP (AX-168291905) located on chromosome 1 exceeded the moderate threshold in both the Additive and Dominant models (Figure 2). This result was supported by the results of the rhAmp SNP genotyping (Table 2, Figure 4). This SNP is located on the intron of the bovine SPATA16 gene and in close proximity to the coding region of GPX6 and ZSCAN23. SPATA16, previously named NYD-SP12, is one of the causative agents of male infertility [30]. A mouse model proved that the deletion of the fourth exon of the SPATA16 gene resulted in infertility due to spermatogenic arrest [31]. SPATA16 is highly conserved across mammals, and the rate of identity between human and bovine SPATA16 is more than 85%. Although the function of bovine SPATA16 has not been fully elucidated, it probably functions similarly to human and mouse SPATA16. Human SPATA16 is specifically expressed in the testis except for weak signals in the pancreas and kidneys [32]. Thus, we need to determine the expression level of bovine SPATA16 in each tissue, especially in BLV-infected female cattle. The effect of this SNP on testis development or spermatogenesis in breeding cattle is not well understood. It is necessary to clarify this point as well, otherwise, this SNP could be connected with other close genes’ expression levels or functions such as GPX6 or ZSCAN23 and affect the PVL.

rhAmp SNP genotyping is based on an RNase H2-dependent PCR combined with a universal reporter system [29]. This high-throughput system enabled us to discriminate the 95 samples of genotypes in one real-time PCR run with a high resolution (Figure 3). Only 2 of 1798 samples were distributed in places outside most of the dots (Figure 1). Thus, we confirmed these genotypes with Sanger sequencing. One sample was determined to be heterozygous with rhAmp SNP genotyping, but the genotype was confirmed as the major homozygous genotype with Sanger sequencing. However, we could not identify why only one sample presented the wrong result because the sequence upstream and downstream of the target SNP was the same as the sequences of most of the other samples. The results of the rhAmp SNP genotyping in the Japanese Black cattle confirmed significant differences in the PVL between cattle with the major homogeneous and the heterogeneous genotypes and between the major and minor homogeneous genotypes. The results show that having a minor allele reduced the PVL by 20–25%. Although a similar tendency to that of the Japanese Black cattle was observed in the Holstein cattle, only a part of the significant difference among the Holstein cattle with each genotype was confirmed. This seems to have been due to the small number of samples from the Holstein cattle.

We focused on an SNP on chromosome 1. However, the Manhattan plots in the Additive and Dominant models showed that many SNPs with low *p*-values were located on chromosome 23 (Figure 2a,b). This tendency was reported in similar previous studies [22,23]. Bovine chromosome 23 contains the BoLA region, and class II is a well-known gene related to BLV resistance and EBL development [13,14,15,16,17,18]. In addition, Takeshima et al. reported that significant SNPs related to the PVL were found in BoLA class I and III regions [22]. Thus, we can conclude that BoLA polymorphisms in coding loci on chromosome 23 are the most influential factors in BLV resistance. However, no similar studies have focused on SNPs on chromosome 1. All other studies have used the Illumina BovineSNP50 chip, while this study used the Axiom Bovine Genotyping Array. This difference could have affected the results of the GWAS because the number and variety of SNPs contained in each array were different.

The results of this study reveal that an untranslated SNP located in the SPATA16-coding region on chromosome 1 is associated with the PVL. However, the mechanisms through which this SNP on SPATA16 impacts the PVL remain unknown. We are considering the possibility that SPATA16 has an unrevealed immunological function or that this SNP affects the expression of other close genes such as GPX6 or ZSCAN23 and affects the PVL. Further functional investigations and replication studies in independent populations are required to corroborate this association with the PVL.

## 5. Conclusions

In this study, we used SNP arrays and performed a GWAS to investigate the entire cattle genome to identify candidate SNPs that could be affected by resistance to BLV. An untranslated SNP (AX-168291905) located on the bovine SPATA16-coding region between exons 4 and 5 was shown to possibly be associated with a low PVL. Then, we confirmed a correlation between the PVL and these genotypes via rhAmp SNP genotyping with an additional 1798 BLV-infected samples. Significant differences in the PVL between the Japanese Black cattle with the major homozygous genotype and the heterozygous genotype and between the Japanese Black cattle with the major and minor homozygous genotypes were confirmed. A similar tendency was also confirmed in the Holstein cattle. Thus, having the minor allele in this SNP would reduce the PVL via an unknown mechanism. Although further studies are needed, this SNP could be a novel biomarker correlated with resistance to BLV.

## Figures and Tables

**Figure 1 vetsci-09-00275-f001:**
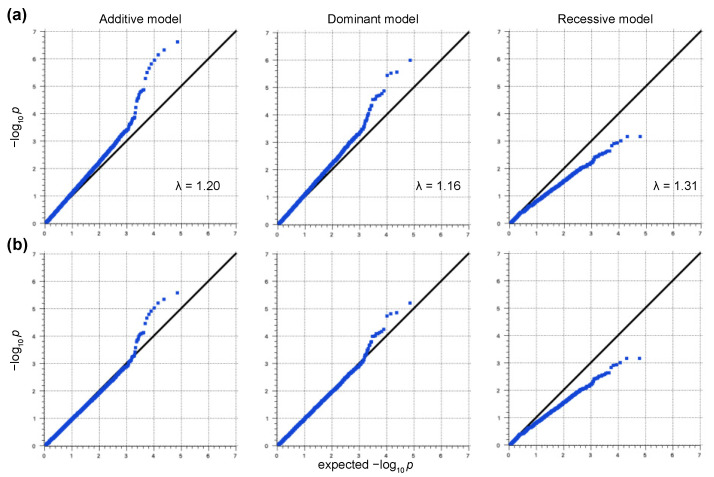
Quantile–quantile (QQ) plots before and after the control of sample stratification. Early deviations of the *p*-values from expected *p*-values suggest population stratifications. The genomic inflation factors (λ) were 1.20, 1.16, and 1.31 in the Additive, Dominant, and Recessive models, respectively (**a**). QQ plots after the control of sample stratification by the genomic control method (**b**).

**Figure 2 vetsci-09-00275-f002:**
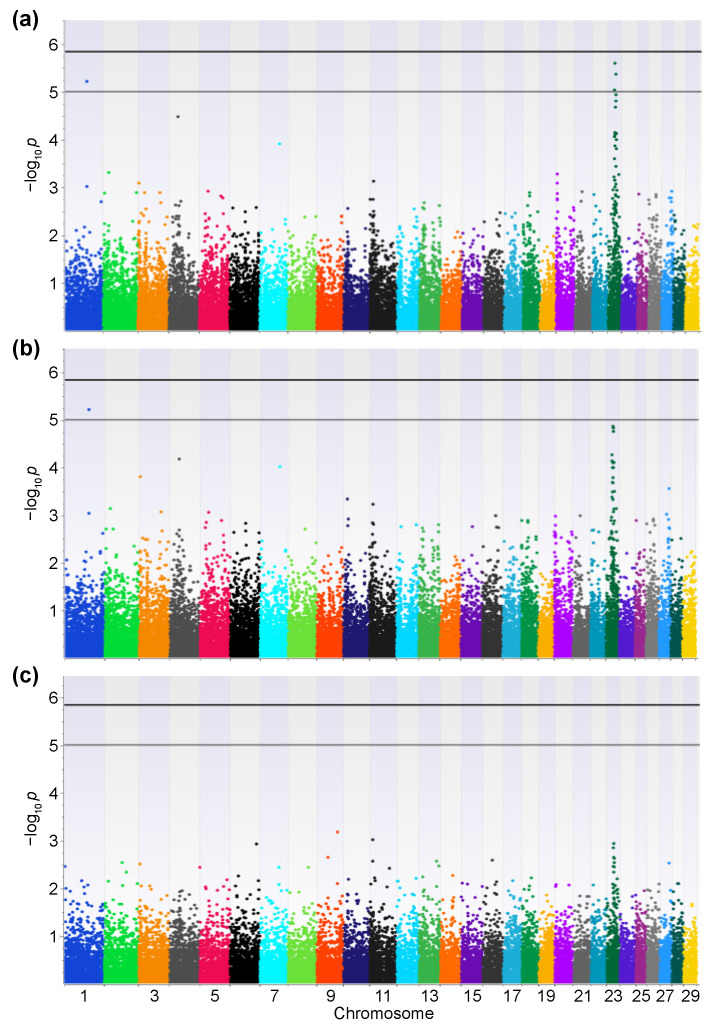
Manhattan plots between cattle with high and low bovine leukemia virus proviral loads. Manhattan plots for the Additive (**a**), Dominant (**b**), and Recessive models (**c**) are depicted. The number of chromosomes and the negative logarithm of the association *p*-values are displayed along the x- and y-axes, respectively. *p*-values were calculated with Fisher’s exact test, and the bold and thin lines indicate the genome-wide significance (*p* = 1.449 × 10^−6^) and the moderate thresholds (*p* = 1.0 × 10^−5^), respectively.

**Figure 3 vetsci-09-00275-f003:**
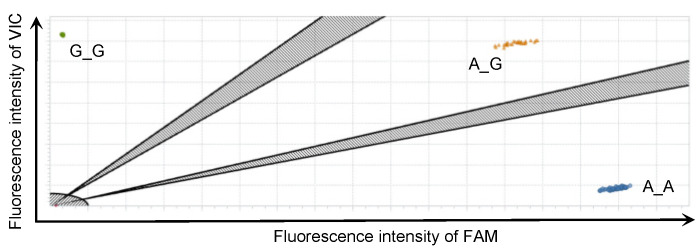
An example of allelic discrimination plots from rhAmp single-nucleotide polymorphism genotyping assay. Allelic discrimination plots for the identified single-nucleotide polymorphism (SNP) (AX-168291905) were obtained using rhAmp SNP genotyping assays on 95 samples with one real-time PCR run. The dots located near the *y*- and *x*-axes represent the minor (G_G) and major (A_A) homozygous genotypes, respectively. The dots located far from the *y*- and *x*-axes represent the heterozygous genotypes (A_G).

**Figure 4 vetsci-09-00275-f004:**
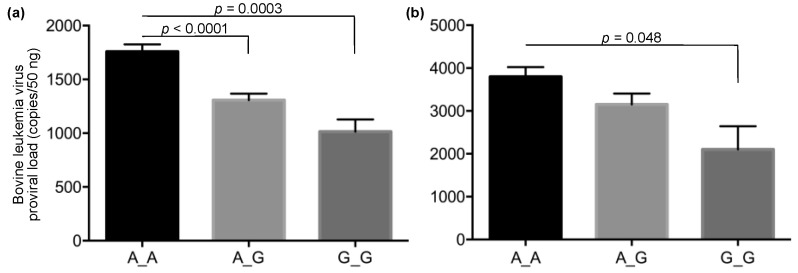
Comparison of bovine leukemia virus proviral load among cattle with each genotype for the identified single-nucleotide polymorphism. Bovine leukemia virus proviral loads (PVLs) were compared among cattle with the major (A_A) and minor (G_G) homozygous and the heterozygous genotypes (A_G) for the identified single-nucleotide polymorphism (SNP) (AX-168291905) in Japanese Black (**a**) and Holstein cattle (**b**). The bars and error bars indicate the means and standard errors of the means, respectively. The median (mean) values of the PVL were 1278 (1757) copies/50 ng for the major homozygous genotype, 843 (1307) copies/50 ng for the heterozygous genotype, and 621 (1015) copies/50 ng for the minor homozygous genotype in Japanese Black cattle (**a**). The median (mean) values of the PVL were 2896 (3799) copies/50 ng for the major homozygous genotype, 2033 (3153) copies/50 ng for the heterozygous genotype, and 388 (2101) copies/50 ng for the minor homozygous genotype in Holstein cattle (**b**).

**Table 1 vetsci-09-00275-t001:** Single-nucleotide polymorphisms (SNPs) with significant difference as a result of genome-wide association study.

SNP ID	Chromosome	Position ^a^	Model	*p*-Value ^b^	Minor Allele	Minor Allele Frequency
AX-168291905	1	95,193,165	Additive	6.15 × 10^−6^	G	0.214
			Dominant	6.03 × 10^−6^		
AX-185116447	23	28,568,745	Additive	2.53 × 10^−6^	A	0.453
AX-115113358	23	30,773,701	Additive	4.34 × 10^−6^	T	0.361
AX-124375550	23	25,642,674	Additive	9.13 × 10^−6^	T	0.383

^a^ SNP location was based on the bovine reference genome assembly Bos taurus UMD3.1. ^b^ The *p*-values were modified by the genomic control method.

**Table 2 vetsci-09-00275-t002:** Proportion of each genotype in identified single-nucleotide polymorphism between high and low bovine leukemia virus proviral load groups.

	Japanese Black Cattle ^b^			Holstein Cattle ^b^		
Genotype	A_A	A_G	G_G	A_A	A_G	G_G
High PVL group ^a^	64.0% (261)	31.4% (128)	4.7% (19)	60.6% (151)	34.9% (87)	4.4% (11)
Low PVL group ^a^	45.5% (112)	44.7% (110)	9.8% (24)	46.5% (46)	44.4% (44)	9.1% (9)

^a^ High PVL: more than 2000 copies/50 ng; Low PVL: fewer than 100. ^b^ Significant differences were observed in Japanese Black (*p* < 0.0001) and Holstein cattle (*p* = 0.031) by the chi-square test.

## Data Availability

The data presented in this study are available on request from the corresponding author. The data are not publicly available due to the need to protect the genetic information of Japanese Black cattle.

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
