# Peer review of "Single-Nucleotide Polymorphism on Spermatogenesis Associated 16 Gene-Coding Region Affecting Bovine Leukemia Virus Proviral Load"

_vetsci, 2022, doi:10.3390/vetsci9060275_

Round 1

Reviewer 1 Report

This is a well written manuscript that provides an interesting insight into identification the novel SNPs related to resistance to BLV. I have some comments.

Line 108-110 : 1798 samples for rhAmp SNP genotyping - How was the range of BLV PVL in this group of animals, what was the proportion of animals in each PVL group?

This is a new gene in this type of study, if we consider resistance - shouldn't there also be a view of SNP profiles of SPATA16  in uninfected animals?

Line 189, Figure 3 - Is it to be understood that this is a random, sample result of the rhAmp SNP genotyping analysis? ( 'just' 95 samples)

Line 227-233 I have some concerns about the typing of this gene for resistance to BLV, and I think the authors do as well, because it is associated with testis development and spermatogenesis.

Line 272-274  ‘ An 271 SNP (AX-168291905) located on the bovine SPATA16-coding region between exons 4 and 272 5 was shown to possibly be associated with a low PVL’    seems to be a conclusion drawn too far - related to low PVL - rather 621 copies for Japanese Black cattle and 388 for Holstein cattle is not  less than 100 copies/50ng – low PVL – please explain

Author Response

Dear Reviewer 1,

Thank you very much for making time in your busy schedule. We are grateful for the reviewer’s valuable comments to improve our manuscript. Our responses are written below. The revised words are marked in red font.

Reviewer 1’s comments and our responses

Comment#1. Line 108-110: 1798 samples for rhAmp SNP genotyping - How was the range of BLV PVL in this group of animals, what was the proportion of animals in each PVL group?

Response#1. Thank you for your comments. We added two sentences as a follow regarding the reviewer’s opinions.

Line 110-113: The median (mean) values of the PVLs in Japanese Black cattle were 1042 (1529) copies/50 ng including 408 cattle of high- and 246 cattle of low-PVL groups. On the other hand, the median (mean) values of the PVLs in Holstein cattle were 2399 (3468) copies/50 ng including 249 cattle of high- and 99 cattle of low-PVL groups.

Comment#2. This is a new gene in this type of study, if we consider resistance - shouldn't there also be a view of SNP profiles of SPATA16 in uninfected animals?

Response#2. Thank you for your comments. Since the values of minor allele frequency are shown in Table 1, we think it is not necessary to check the SNP genotype in uninfected cattle. Furthermore, as pointed out by another reviewer, Table 2 has been added. We think this table is also be helpful to understand.

Comment#3. Line 189, Figure 3: Is it to be understood that this is a random, sample result of the rhAmp SNP genotyping analysis? ( 'just' 95 samples)

Response#3. Yes, we showed a result of one real-time PCR run. To avoid being confused, we add “an example of” in Figure title as follows.

Line 197: Figure 3. An example of allelic discrimination plots from rhAmp single-nucleotide polymorphism genotyping assay.

Comment#4. Line 227-233: I have some concerns about the typing of this gene for resistance to BLV, and I think the authors do as well, because it is associated with testis development and spermatogenesis.

Response#4. Thank you for your comments, we added the two sentences as a follow regarding the reviewer’s opinions.

Line 246- 247: The effect of this SNP on testis development or spermatogenesis in breeding cattle is not well understood. It is necessary to clarify this point as well.

Comment#5. Line 272-274: ‘An 271 SNP (AX-168291905) located on the bovine SPATA16-coding region between exons 4 and 5 was shown to possibly be associated with a low PVL’ seems to be a conclusion drawn too far - related to low PVL - rather 621 copies for Japanese Black cattle and 388 for Holstein cattle is not less than 100 copies/50ng – low PVL – please explain.

Response#5. Thank you for your comments. This sentence indicates only the result of GWAS not rhAmp genotyping. As pointed out by another reviewer, Table 2 has been added. This table is also be helpful to understand the result of rhAmp genotyping.

Reviewer 2 Report

Mekata and Yamamoto found in this study that a SNP on SPATA16-coding region is associated with BLV proviral load. This result is very interesting. However, the study has one critical problem as follows:

(1) In the Materials and Methods section, it is described that the authors classified cattle with more than 2000 and fewer than 100 copies/50 ng as high- and low-PVL groups, respectively. However, in Figure 4, these is no cattle with fewer than 100 copies/50 ng. So, the SNP is associated with PVL only in the high-PVL group. Does the low-PVL group have higher rate of G_G than the high-PVL group? This issue must be clarified.

Minor comment

(2) The SNP is located on the coding region of SPATA16 gene. Does the SNP change the amino acid?

(3) SPATA16 gene is specifically expressed in testis, and thought to be involved in spermatogenesis. Is the SNP associated with sexual transmission of BLV? (I do not know whether BLV is sexually transmitted). Please discuss how the SNP affects PVL.

Author Response

Dear Reviewer 2,

Thank you very much for making time in your busy schedule. We are grateful for your valuable comments to improve our manuscript. Our responses are written below. The revised words are marked in red font.

Reviewer 1’s comments and our responses

Comment#1. In the Materials and Methods section, it is described that the authors classified cattle with more than 2000 and fewer than 100 copies/50 ng as high- and low-PVL groups, respectively. However, in Figure 4, these is no cattle with fewer than 100 copies/50 ng. So, the SNP is associated with PVL only in the high-PVL group. Does the low-PVL group have higher rate of G_G than the high-PVL group? This issue must be clarified.

Response#1. Thank you for your comment. We have added Table 2 regarding the reviewer's suggestions. In addition to that, we added some sentences as follows including statistical analysis.

Line 204-208: A table 2 was added.

Line 180-183: In the high PVL group, 4.7% (19/408) of Japanese Black and 4.4% (11/249) of Holstein had the minor (G_G) homozygous genotype. Whereas, 9.8% (11/249) of Japanese Black and 9.1% (9/99) of Holstein had this genotype in the low PVL group (p<0.0001).

Line 130-131: The Chi-square test was used for comparisons between the coloration of PVL groups and each genotype.

Comment#2. The SNP is located on the coding region of SPATA16 gene. Does the SNP change the amino acid?

Response#2. Thank you for your comment. This SNP is located in the untranslated region. To avoid confusion, we revised the words “SNP” to “untranslated SNP”.

Line 279: The results of this study reveal that an untranslated SNP located in the SPATA16-

Line 289: An untranslated SNP (AX-168291905) located on…

Comment#3. SPATA16 gene is specifically expressed in testis, and thought to be involved in spermatogenesis. Is the SNP associated with sexual transmission of BLV? (I do not know whether BLV is sexually transmitted). Please discuss how the SNP affects PVL.

Response#3. Thank you for your comments. We added two sentences as a follow regarding the reviewer’s opinions.

Line 281-284: We are considering the possibility that SPATA16 has an unrevealed immunological function or that this SNP affects the expression of other close genes such as GPX6 or ZSCAN23 and affects the PVL.

Bovine leukemia virus could be transmitted by mating. However, artificial insemination is common in the cattle industry, the semen used for artificial insemination is frozen, and the virus is inactivated by frozen. In general, non-BLV infected cows can be a sire. Thus, we think this SNP does not affect the sexual transmission of BLV.

Round 2

Reviewer 1 Report

Thanks to the authors for responding to the suggestions made to their article.

Reviewer 2 Report

The manuscript was correctly revised.